# Large-scale whole genome sequencing of *M. tuberculosis* provides insights into transmission in a high prevalence area

JA Guerra-Assunção[1], AC Crampin[1,2], RMGJ Houben[1], T Mzembe[2], K Mallard[3], F Coll[3], P Khan[1], L Banda[2], A Chiwaya[2], RPA Pereira[3], R McNerney[3], PEM Fine[1], J Parkhill[4], TG Clark[3], JR Glynn[1]*

[1]Faculty of Epidemiology and Population Health, London School of Hygiene and Tropical Medicine, London, United Kingdom; [2]Karonga Prevention Study, Malawi, Malawi; [3]Faculty of Infectious and Tropical Diseases, London School of Hygiene and Tropical Medicine, London, United Kingdom; [4]Wellcome Trust Sanger Institute, Hinxton, United Kingdom

**Abstract** To improve understanding of the factors influencing tuberculosis transmission and the role of pathogen variation, we sequenced all available specimens from patients diagnosed over 15 years in a whole district in Malawi. *Mycobacterium tuberculosis* lineages were assigned and transmission networks constructed, allowing ≤10 single nucleotide polymorphisms (SNPs) difference. We defined disease as due to recent infection if the network-determined source was within 5 years, and assessed transmissibility from forward transmissions resulting in disease. High-quality sequences were available for 1687 disease episodes (72% of all culture-positive episodes): 66% of patients linked to at least one other patient. The between-patient mutation rate was 0.26 SNPs/year (95% CI 0.21–0.31). We showed striking differences by lineage in the proportion of disease due to recent transmission and in transmissibility (highest for lineage-2 and lowest for lineage-1) that were not confounded by immigration, HIV status or drug resistance. Transmissions resulting in disease decreased markedly over time.

*For correspondence: judith. glynn@lshtm.ac.uk

**Competing interests:** The authors declare that no competing interests exist.

**Reviewing editor**: Quarraisha Abdool Karim, University of KwaZulu Natal, South Africa

## Introduction

Despite the huge global burden of tuberculosis, the factors influencing transmission remain poorly understood. Compared to other bacteria, the genome of *Mycobacterium tuberculosis* is stable and genetic variation was thought to be limited, but with increased sequencing, greater diversity has been recognized (*Homolka et al., 2010*). Based on the genotype, *M. tuberculosis* has seven lineages: three 'ancient' (lineage-1 and two *Mycobacterium africanum* lineages), and three 'modern' (lineages-2, 3, 4) (*Comas et al., 2009*), and one intermediate (lineage-7), recently described in Ethiopia (*Firdessa et al., 2013*). The lineages may vary in propensity to transmit and cause disease (*Thwaites et al., 2008*; *Homolka et al., 2010*; *Parwati et al., 2010*; *Gagneux, 2012*), but results are inconsistent and there is considerable strain-to-strain variation within lineages (*Portevin et al., 2011*; *Mathema et al., 2012*).

Lineage-2 (Beijing) strains are associated with increasing spread and drug resistance in some areas but not others (European Concerted Action on New Generation Genetic Markers, 2006), and with a lower (*Click et al., 2012*) or higher (*Kong et al., 2007*) proportion of extrapulmonary tuberculosis. *M. africanum* has been associated with lower virulence (*de Jong et al., 2008*), and lineage-1 with faster sputum smear conversion (*Click et al., 2013*). In low incidence settings, lineage is often associated with immigrant sub-groups, and while host–pathogen co-evolution has been suggested, it is difficult to disentangle the effects of lineage and host susceptibility on pathogenesis (*Reed et al., 2009*; *Gagneux, 2012*; *Pareek et al., 2013*).

**eLife digest** Tuberculosis is an important public health threat around the globe and is particularly common in developing countries. It is difficult to control the spread of the disease because the bacteria that cause it can spread when an infected individual coughs or sneezes. It may take years for an infected individual to develop symptoms of tuberculosis so it can be hard to trace the source of an outbreak, and people infected with HIV are particularly susceptible to the disease.

The bacterium that causes the majority of cases of tuberculosis is called *Mycobacterium tuberculosis*. There are several different varieties or 'lineages' of *M. tuberculosis*, and it is thought that they may vary in their ability to spread and cause disease. However, the results of previous studies have been inconsistent and there also seems to be a lot of variation between strains within the same lineage.

In this study, Guerra-Assunção et al. used an approach called whole genome sequencing alongside more traditional methods to study the spread of tuberculosis in Malawi. They sequenced the genomes of every available sample of *M. tuberculosis* collected from patients in the Karonga district of Malawi over a 15-year period. This produced high-quality DNA sequence data about the bacteria responsible for almost 1700 cases of disease.

Using this massive amount of data, Guerra-Assunção et al. constructed networks that showed how the bacteria had spread in the community. This revealed that there were differences between the ability of the various *M. tuberculosis* lineages to cause disease and to spread in communities. For example, lineage 1 was less likely than the other lineages to cause disease soon after infecting an individual and was less able to spread.

The data also show that the proportion of cases of disease due to recent infection declined substantially during the 15-year period. This indicates that the tuberculosis and HIV control programmes in the area have been successful.

Guerra-Assunção et al.'s findings show that it is possible to understand how tuberculosis is transmitted on a large scale. The next challenge is to understand why the lineages differ in their ability to cause disease and spread between individuals.

Since the 1990s, methods such as RFLP based on the insertion element IS6110 (*van Embden et al., 1993*) have been used to distinguish clusters of patients with shared DNA-fingerprint patterns, suggesting recent transmission (*Small et al., 1994*), but within the clusters, these methods cannot distinguish who transmitted to whom. Whole genome sequencing provides far greater resolution, and if data are collected in a whole population over several years, single nucleotide polymorphisms (SNPs) can be used to construct transmission networks (*Bryant et al., 2013*; *Walker et al., 2013*, *2014*). In low-incidence settings small numbers of SNPs have been found between epidemiologically linked patients (*Kato-Maeda et al., 2013*), although the maximum SNP difference to 'confirm' a link is not yet established (*Perez-Lago et al., 2014*). No population-based study to-date has applied long-term large-scale whole genome sequencing in a high prevalence area (*Luo et al., 2014*; *Walker et al., 2014*), it is much more challenging to interpret transmission networks when there are many possible sources of infection. Yet understanding transmission in high prevalence areas would have the greatest public health benefit.

As part of the Karonga Prevention Study in Malawi, we assess transmission using whole genome sequencing in the whole district over 15 years. We show decreasing transmission over time and marked variation between *M. tuberculosis* lineages 1–4 which are unconfounded by host differences.

## Results

Between September 1995 and September 2010, there were 2332 person-episodes of culture-confirmed tuberculosis in Karonga District. Whole genome sequences that passed quality control were available for 1687 (72%). The distribution of patients with and without sequences available was very similar by age, sex, and HIV status. The proportion with sequences available was the highest in 2002–2006 (82%) and was higher in those with smear-negative pulmonary disease (78%) than in those with smear-positive disease (71%) and extra-pulmonary disease (67%).

The phylogenetic tree is shown in *Figure 1*. Most *M. tuberculosis* strains (68%) were lineage-4, with 16% lineage-1, 4% lineage-2, and 12% lineage-3 (*Table 1*). Lineage-4 strains were more common in the earlier years. Lineage-1 strains were more common in HIV-positive and older patients and less common in recurrent tuberculosis. Lineage-2 strains were more common in younger patients and were all drug sensitive. Lineage-3 strains were associated with recurrent tuberculosis and with isoniazid resistance. There was no association between lineage and having been born or recently resident

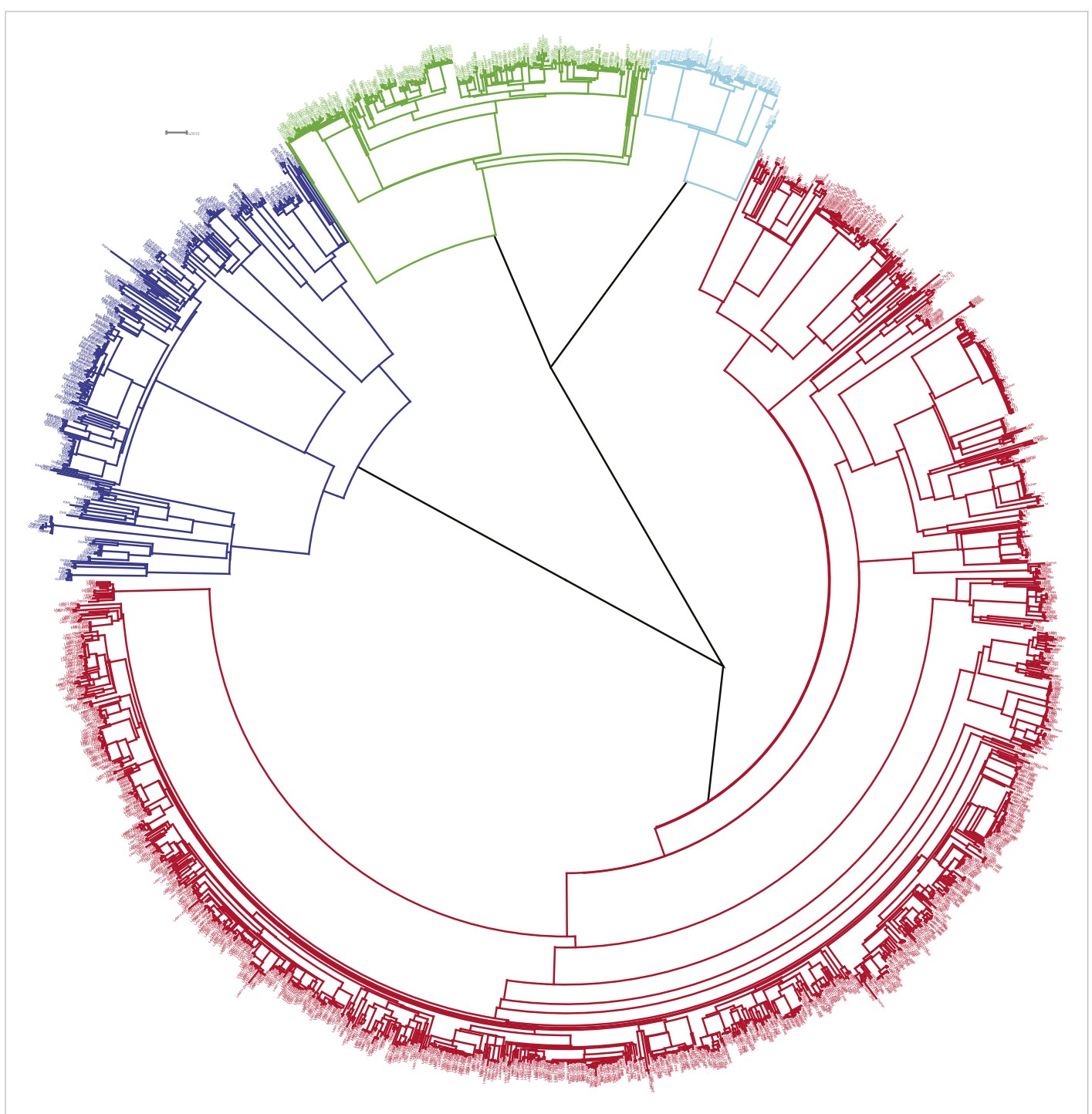

**Figure 1**. Phylogenetic tree of all samples from Karonga. Lineages form monophyletic groups within the phylogeny, as expected. Lineage 1 (Indo Oceanic) is represented in dark blue, Lineage 2 (Beijing/East Asian) in light blue, Lineage 3 (East African Indian) in green, and Lineage 4 (Europe American) in red.

**Table 1.** Characteristics of patients included in the analysis and distribution of lineages

| | Lineage | | | | | |
| | 1 | 2 | 3 | 4 | Overall | p* |
|---|---|---|---|---|---|---|
| Overall | 269 (16.0) | 74 (4.4) | 205 (12.2) | 1139 (67.5) | 1687 | |
| **Age** | | | | | | |
| <20 | 9 (12.3) | 7 (9.6) | 9 (12.3) | 48 (65.7) | 73 | |
| 20–29 | 46 (10.3) | 26 (5.8) | 48 (10.7) | 327 (73.2) | 447 | |
| 30–39 | 109 (18.4) | 17 (2.9) | 81 (13.7) | 386 (65.1) | 593 | |
| 40–49 | 61 (19.8) | 18 (5.8) | 39 (12.7) | 190 (61.7) | 308 | |
| 50+ | 44 (16.5) | 6 (2.3) | 28 (10.5) | 188 (70.7) | 266 | 0.001 |
| **Sex** | | | | | | |
| Female | 130 (14.6) | 47 (5.3) | 94 (10.6) | 617 (69.5) | 888 | |
| Male | 139 (17.4) | 27 (3.4) | 111 (13.9) | 522 (65.3) | 799 | 0.02 |
| **Year** | | | | | | |
| 1995–1998 | 55 (15.5) | 8 (2.3) | 29 (8.2) | 263 (74.1) | 355 | |
| 1999–2001 | 43 (11.5) | 23 (6.1) | 43 (11.5) | 266 (70.9) | 375 | |
| 2002–2004 | 80 (19.4) | 22 (5.3) | 54 (13.1) | 257 (62.2) | 413 | |
| 2005–2007 | 54 (17.4) | 11 (3.5) | 44 (14.2) | 202 (65.0) | 311 | |
| 2008–2010 | 37 (15.9) | 10 (4.3) | 35 (15.0) | 151 (64.8) | 233 | 0.004 |
| **TB type** | | | | | | |
| Smear+ | 212 (17.3) | 52 (4.3) | 156 (12.8) | 804 (65.7) | 1224 | |
| Smear− | 46 (12.1) | 19 (5.0) | 38 (10.0) | 276 (72.8) | 379 | |
| Extrapulmonary | 11 (13.1) | 3 (3.6) | 11 (13.1) | 59 (70.2) | 84 | 0.1 |
| **HIV status** | | | | | | |
| Negative | 47 (10.8) | 23 (5.3) | 57 (13.0) | 310 (70.9) | 437 | |
| Positive | 148 (19.3) | 28 (3.6) | 107 (13.9) | 486 (63.2) | 769 | 0.001 |
| **Previous TB** | | | | | | |
| No | 251 (16.7) | 66 (4.4) | 171 (11.4) | 1019 (67.6) | 1507 | |
| Yes | 18 (10.0) | 8 (4.4) | 34 (18.9) | 120 (66.7) | 180 | 0.007 |
| **Isoniazid resistance** | | | | | | |
| Resistant | 20 (17.2) | 0 (0.0) | 21 (18.1) | 75 (64.7) | 116 | |
| Sensitive | 244 (15.9) | 74 (4.8) | 181 (11.8) | 1033 (67.4) | 1532 | 0.03 |
| **Residence** | | | | | | |
| Karonga | 198 (16.4) | 53 (4.4) | 148 (12.3) | 806 (66.9) | 1205 | |
| Malawi | 48 (16.6) | 13 (4.5) | 32 (11.1) | 196 (67.8) | 289 | |
| Other country | 11 (11.5) | 7 (7.3) | 17 (17.7) | 61 (63.5) | 96 | 0.4 |
| **Birth place** | | | | | | |
| Karonga | 174 (17.0) | 46 (4.5) | 135 (13.2) | 667 (65.3) | 1022 | |
| Malawi | 55 (16.3) | 14 (4.1) | 31 (9.2) | 238 (70.4) | 338 | |
| Other country | 34 (11.7) | 14 (4.8) | 37 (12.7) | 206 (70.8) | 291 | 0.2 |

*From $X^2$ comparison between lineages.

outside the district. The associations of lineage with HIV status and recurrent tuberculosis persisted after adjusting for age, sex, and year. The association between lineage and recurrent tuberculosis was also present when restricted to those with drug-sensitive strains, and the association between lineage and isoniazid resistance was also present when restricted to those with first episode tuberculosis.

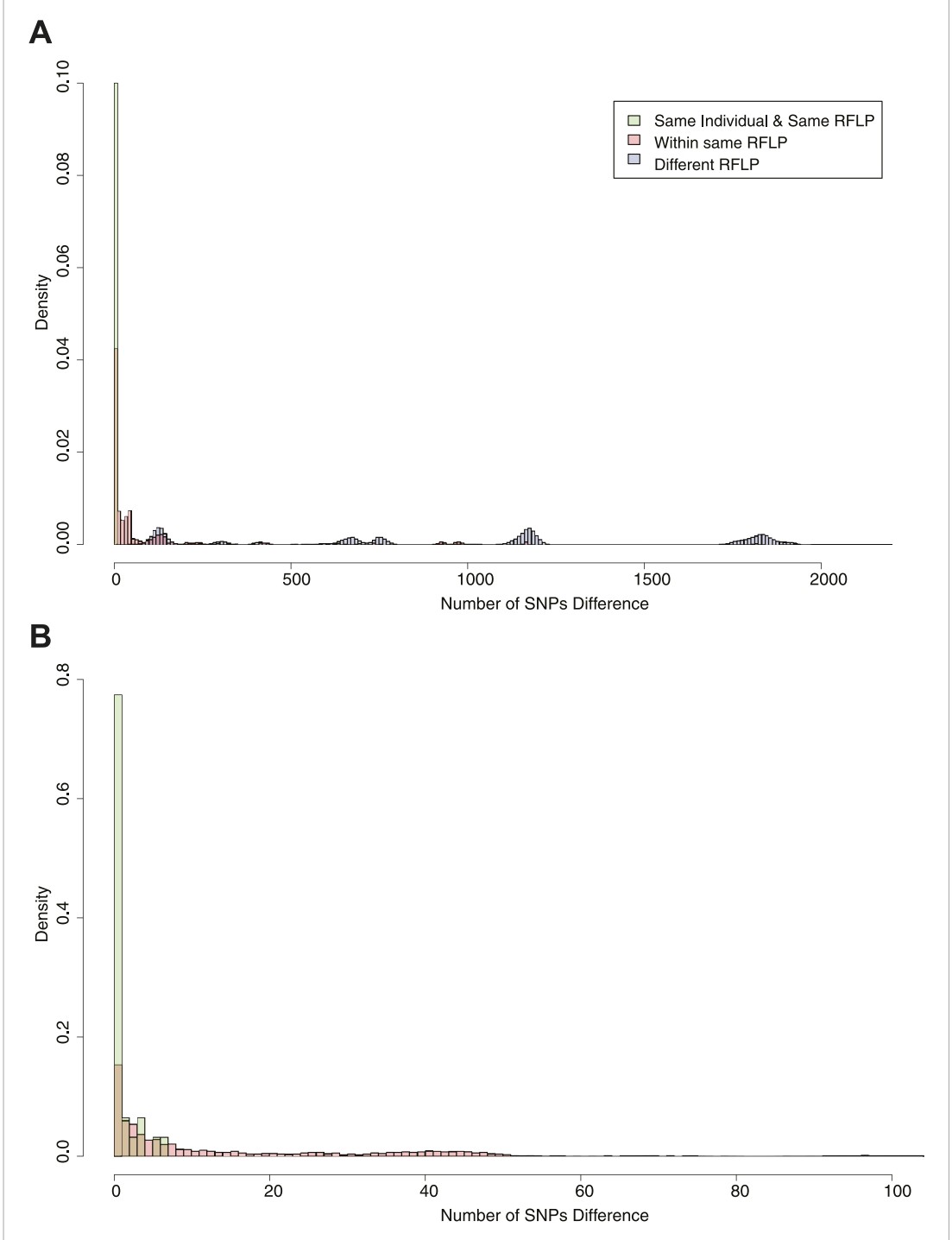

**Figure 2**. Pairwise SNP distances between all pairs of samples with known RFLP. The y axis shows the relative frequency within each subgroup: same RFLP pattern (red), different RFLP patterns (blue); same individual, same RFLP (green). (**A**) shows the full data set, and (**B**) is part of the same figure drawn at a larger scale (each bar corresponds to 1 SNP) to show the smaller distances more clearly.

The following figure supplement is available for figure 2:

**Figure supplement 1**. Pairwise mutation rates between all pairs of samples with known RFLP (calculated as number of SNPs/number of days between dates of disease onset between individuals).

## SNP-based linkage thresholds

*Figure 2* shows the SNP distances between all possible pairs of samples in the data set (including more than one per individual in some cases). Peaks corresponding to large numbers of SNPs represent comparisons between lineages. On the basis of the distribution, we chose cut-offs at 5 and 10 SNPs for distinguishing links. Similar figures were drawn for the mutation rate (*Figure 2—figure supplement 1*). We have previously shown that patients with relapse had up to 8 SNPs difference (*Guerra-Assuncao et al., 2014*), and these cut-offs are similar to those used in other studies (*Bryant et al., 2013*; *Walker et al., 2013*).

## Transmission network

To construct the transmission network, we included links of up to 10 SNPs difference. We included one sample per person-episode of disease and excluded extra-pulmonary cases as they cannot transmit. Example clusters are shown in *Figure 3*, and the full transmission network in *Figure 3—figure supplement 1*. Overall, after excluding relapses (recurrences with ≤10 SNPs difference from the initial episode), 66% of patients were in clusters with at least one other patient. Clusters ranged in size from 2 to 36 (*Figure 4A*), with 23% of patients in clusters of 10 or more. The size of the clusters varied by lineage (*Figure 4B*): compared to lineage-4 (the commonest lineage), lineage-2 and lineage-3 strains were more likely to be clustered and in larger clusters and lineage-1 strains were less likely to be clustered and were in smaller clusters. The median cluster size and interquartile range (IQR) for lineages 1–4 were 3 (1, 6), 13 (7, 24), 7 (2, 22), and 3 (1, 8), respectively. The p-values for differences between lineages were similar if non-clustered strains were excluded.

## Mutation rates

Overall, of 824 links with 0–10 SNPs identified in the networks, 255 (31%) had 0 SNPs different, 182 (22%) had 1 SNP, 127 (15%) had 2 SNPs, 77 (9%) had 3 SNPs, 52 (6%) had 4 SNPs, 32 (4%) had 5 SNPs, and 99 (12%) had 6–10 SNPs different. The number of SNPs correlated with the time between disease onset in the pairs of individuals linked in the network (*Figure 4C*): linear regression $r^2 = 10\%$, $p < 0.001$. The regression coefficient suggests a mutation rate of 0.26 SNPs/year (95% CI 0.21–0.31). The regression results were the same if sputum collection dates were used instead of disease onset dates.

The within-patient mutation rate was calculated in 74 individuals with multiple specimens, including 51 relapses, allowing ≤10 SNPs, and using the first and last specimens if there were more than two. The estimated mutation rate was 0.45 SNPs/year (95% CI 0.15–0.75), $r^2 = 11\%$, $p = 0.004$ (*Figure 4—figure supplement 1*).

*Figure 4D* shows the number of SNPs in the likely transmissions identified from the network, by lineage. Lineage-2 had the lowest number of SNPs per transmission, and lineage 1 the highest. The median mutation rates per year for the different lineages were lineage-1, 0.58 (IQR 0.11–1.9); lineage-2, 0.11 (0–0.66); lineage-3, 0.35 (0–1.1); lineage-4, 0.40 (0–1.2) (p = 0.004, equality-of-medians test). The regression of number of SNPs by number of days showed no clear differences between lineages (*Figure 4—figure supplement 2*).

We investigated the number of SNPs in the likely transmissions by smear status, HIV status, and isoniazid resistance of the initial and subsequent cases. There were no differences by the characteristics of the first case, but transmissions to smear-positive subsequent cases had slightly more SNPs than those to smear-negative subsequent cases (p = 0.05); and those to HIV-negative subsequent cases had slightly more SNPs than those to HIV positive subsequent cases (p = 0.02). Using mutation rates, the results were similar, but with smaller differences by smear status and HIV status of the subsequent case (p = 0.06 and 0.08, respectively).

For further analysis of transmission, we excluded 77 uncertain links (i.e., with 6–10 SNPs and mutation rate ≥0.003 SNPS/day, *Figure 2—figure supplement 1*).

## Recent infection

A case of tuberculosis was defined as being due to recent infection if a source case was identified in the network within the previous 5 years, and not being due to recent infection if no source was identified or if the closest source (in terms of number of SNPs) was more than 5 years earlier. Overall,

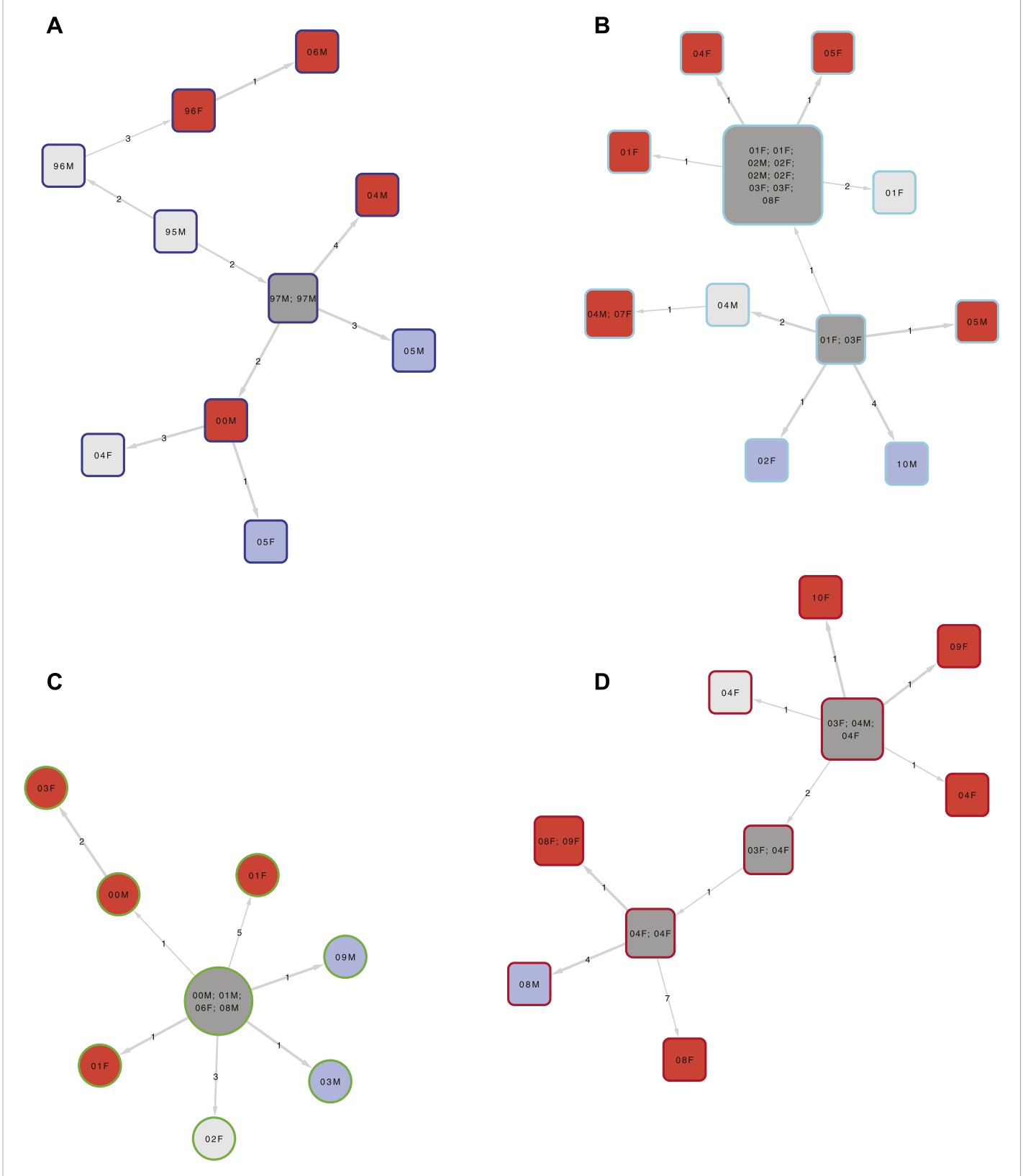

**Figure 3**. Examples of clusters built using SeqTrack. All clusters are shown in *Figure 3—figure supplement 1*. Each polygon represents a patient, with larger polygons representing two or more patients with identical sequences. The patient details are written inside the polygon: F = female, M = male.
*Figure 3. continued on next page*

*Figure 3. Continued*
The number is the year of the start of the disease episode. The shapes describe drug resistance of the strain: squares = drug sensitive, circles = drug resistant. The colour of the polygon refers to HIV status of the patient: red = positive, blue = negative, grey = unknown (or multiple patients). The colour of the edge refers to the lineage: Lineage 1 (Indo Oceanic) dark blue (**B**), Lineage 2 (Beijing/East Asian) light blue (**C**), Lineage 3 (East African Indian) green (**A**), and Lineage 4 (Europe American) red (**D**). The numbers on the arrows between the polygons are the number of SNPs between them.
The following figure supplement is available for figure 3:

**Figure supplement 1**. Clusters built using SeqTrack.

38% of patients had evidence of recent infection (*Table 2*). This was the highest for lineage-2 (65%) and the lowest for lineage-1 (31%). Linkage with a recent source case was less common in older age groups, in those who had been living outside the district, and in more recent years, with the proportion linked decreasing from 45% in 1999–2001 to 30% in 2008–2010. These trends persisted after adjusting for each other (*Table 2*). There was no association of linkage with sex, HIV status, sputum smear status, or isoniazid resistance and adjusting for these did not affect the results. The effect of village of birth was lost after adjusting for recent residence.

## Transmissibility

From the network, 32% of individuals were linked as likely sources of infection to at least one other individual. Individuals were sources for up to 12 others, with 293 (22%) linked to one, 76 (6%) linked to two, 22 (2%) linked to three, 14 (1%) linked to four, and 26 (2%) linked to five or more.

*Table 3* shows the association of characteristics of the index episode with the likelihood of transmission, using ordered logistic regression. There were more transmissions from those with positive smears, and with tuberculosis in the earlier years. Lineage-2 and lineage-3 strains were more likely to transmit than lineage-4, and these differences were more marked after adjustment for year, age, sex, and smear status. Place of birth and recent residence were weakly associated with onward transmission, and further adjusting for these or the other factors in the table did not affect the results.

Comparing those with any transmissions vs those with none in a logistic regression model gave very similar results (not shown). Restricting the links to those within 3 years of the index episode, there was still a strong trend with year: the odds ratios from the ordered logistic regression analysis, adjusted for lineage, age, sex, and smear status, for the year groups 1999–2001, 2002–2004 and 2005–2007, compared to 1995–1998, were 0.47 (95% CI 0.33–0.66), 0.35 (0.25–0.50), and 0.37 (0.25–0.54), respectively.

## Discussion

This is the largest whole genome sequencing study of *M. tuberculosis* transmission to-date, and the first to use a network approach. We show that this approach is feasible and that with long-term, population-wide data, important inferences can be made about transmission. In this population, although lineage-4 has been present for longer (*Glynn et al., 2010*), lineages are not now associated with area of birth or recent residence, so differences by lineage are unlikely to be confounded by associations with host sub-populations.

The mutation rates in this study are consistent with those from other settings (*Bryant et al., 2013*; *Walker et al., 2013*) and in vitro (*Ford et al., 2013*). This is the largest study to measure between-patient mutation rates. Although the confidence intervals on the estimate are narrow, there is considerable variation as others have found. The measure assumes the correct source has been identified and uses the time interval between dates of when the disease was first diagnosed or specimen collection as necessarily crude approximations of the time since divergence of the samples from their common ancestor. Furthermore, there is a bottleneck on transmission: most infections probably arise from one or very few organisms (*Lurie, 1964*), which may be minority strains in the first case. The within-patient estimate of mutation rate does not have the same measurement problems and gave a consistent result.

We showed striking differences between the lineages in cluster size, the proportion of disease due to recent transmission and transmissibility. Lineage-1 formed the smallest clusters, with the largest

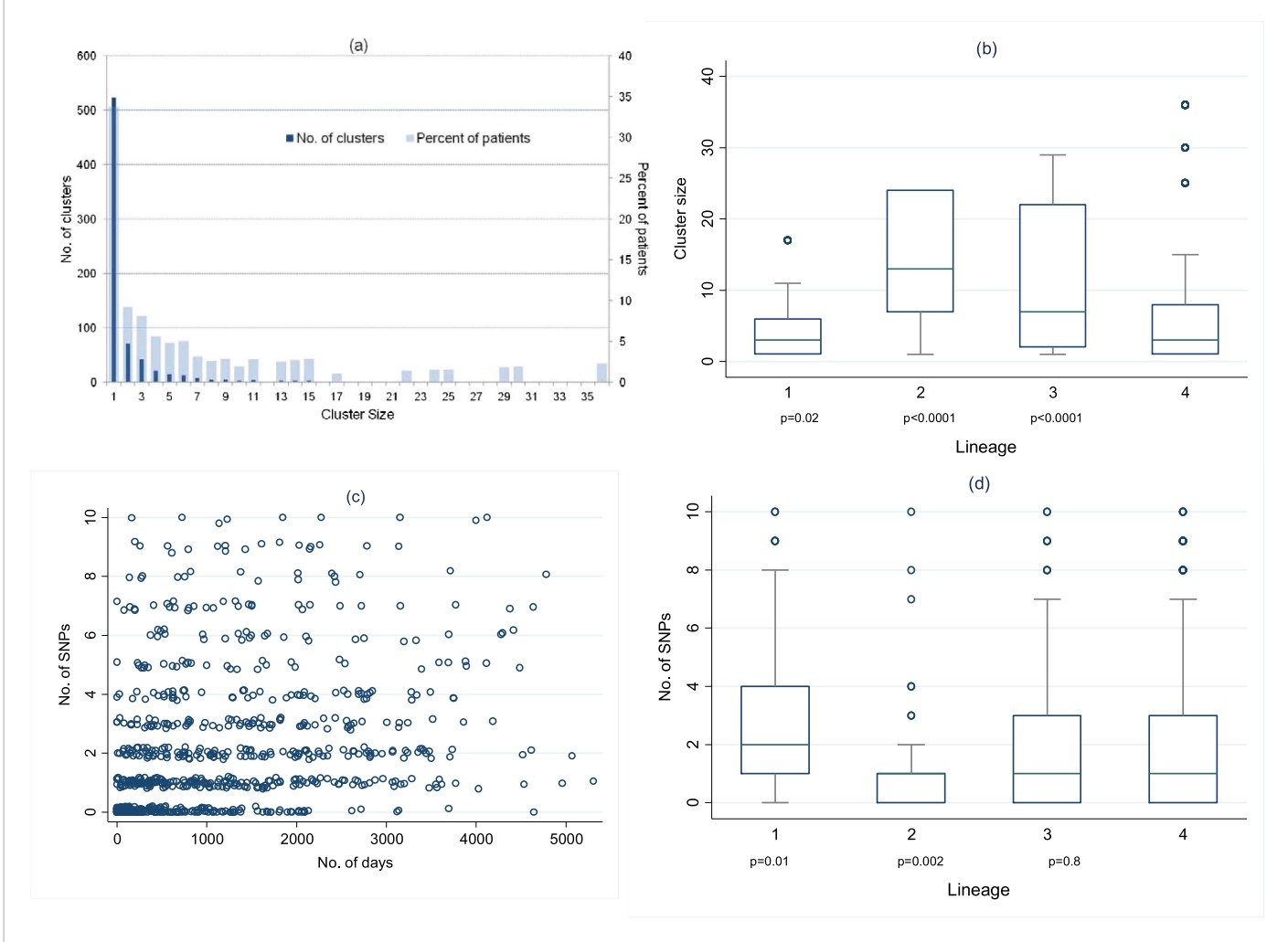

**Figure 4**. Distribution of clusters and SNPs. (**A**) Number of clusters of different sizes and percentage of patients in clusters of different sizes. Cluster size 1 refers to unclustered patients. (**B**) Cluster size by lineage. The p values are for the comparison of each lineage with lineage-4 (Wilcoxon rank sum test). (**C**) Relationship between number of SNPs between individuals and the time interval between disease onset in each individual of the pair. (Random noise has been introduced to allow multiple similar results to be visualized.) Linear regression gives $r^2 = 10\%$, $p < 0.001$, slope 0.26 SNPs per year (95% CI 0.21–0.31). (**D**) Number of SNPs between individuals in clusters, by lineage. The p values are for the comparison of each lineage with lineage-4 (Wilcoxon rank sum test).

The following figure supplements are available for figure 4:

**Figure supplement 1**. Relationship between number of SNPs and the number of days between samples from individuals with more than one specimen available from the same of episode of disease or from a relapse.

**Figure supplement 2**. Relationship between number of SNPs and the number of days between dates of disease onset for transmissions identified from the network, by lineage.

SNP differences. Patients with lineage-1 strains were the least likely to have disease due to recent transmission and were less likely to transmit and cause new cases than those with lineages 2 or 3. These observations suggest a lower propensity to cause disease, which may explain lineage-1's association with HIV infection, if it is less likely to cause active disease in those who are not immunosuppressed. Lineage-1 strains have been associated with lower virulence in animal models (**Narayanan et al., 2008**; **Reiling et al., 2013**).

**Table 2**. Characteristics associated with disease due to recent infection

| Characteristic | Linked/Total n/N | % | Association with links (unadjusted) OR (95% CI) | p (lrtest) | Adjusted for age, sex, year, lineage OR (95% CI) | Adjusted for other variables included in model* OR (95% CI) | p (lrtest) |
|---|---|---|---|---|---|---|---|
| Overall | 409/1074 | 38.1 | | | | | |
| Lineage | | | | | | | |
| 1 | 56/183 | 30.6 | 0.76 (0.53–1.1) | | 0.81 (0.57–1.2) | 0.81 (0.57–1.2) | |
| 2 | 34/52 | 65.4 | 3.2 (1.8–5.9) | | 3.0 (1.6–5.4) | 3.2 (1.7–5.8) | |
| 3 | 58/129 | 45.0 | 1.4 (0.96–2.1) | | 1.5 (1.0–2.2) | 1.5 (1.0–2.2) | |
| 4 | 261/710 | 36.8 | 1 | <0.001 | 1 | 1 | <0.001 |
| Age | | | | | | | |
| <20 | 19/36 | 65.8 | 2.9 (1.4–6.0) | | 2.5 (1.2–5.4) | 2.6 (1.2–5.6) | |
| 20–29 | 113/276 | 45.8 | 1.8 (1.2–2.7) | | 1.6 (1.1–2.5) | 1.8 (1.2–2.8) | |
| 30–39 | 152/404 | 39.6 | 1.5 (1.0–2.3) | | 1.5 (0.99–2.2) | 1.6 (1.0–2.3) | |
| 40–49 | 81/201 | 44.2 | 1.7 (1.1–2.7) | | 1.0 (1.0–2.6) | 1.7 (1.1–2.6) | |
| 50+ | 44/157 | 33.5 | 1 | 0.007† | 1 | 1 | 0.03† |
| Sex | | | | | | | |
| Female | 229/575 | 39.8 | 1 | | | | |
| Male | 180/499 | 36.1 | 0.85 (0.67–1.1) | 0.05 | 0.93 (0.72–1.2) | 0.94 (0.72–1.2) | 0.4 |
| Year | | | | | | | |
| 1999–2001 | 141/311 | 45.3 | 1 | | 1 | 1 | <0.001† |
| 2002–2004 | 117/322 | 36.3 | 0.69 (0.50–0.95) | | 0.73 (0.52–1.0) | 0.69 (0.50–0.97) | |
| 2005–2007 | 92/244 | 37.7 | 0.73 (0.52–1.0) | | 0.78 (0.55–1.1) | 0.70 (0.49–1.0) | |
| 2008–2010 | 59/197 | 30.0 | 0.52 (0.35–0.75) | 0.001† | 0.53 (0.36–0.77) | 0.48 (0.32–0.70) | |
| TB type | | | | | | | |
| Smear-positive pulmonary | 312/821 | 38.0 | 1 | | 1 | | |
| Smear-negative pulmonary | 97/253 | 38.3 | 1.0 (0.76–1.4) | 0.9 | 0.95 (0.71–1.3) | | |
| HIV status | | | | | | | |
| HIV– | 102/283 | 36.0 | 1 | | | | |
| HIV+ no ART | 173/436 | 39.7 | 1.2 (0.85–1.6) | | 1.1 (0.75–1.5) | | |
| HIV+ on ART | 27/77 | 35.1 | 0.96 (0.56–1.6) | 0.5 | 1.0 (0.56–1.8) | | |
| INH resistance | | | | | | | |
| No | 375/979 | 38.3 | 1 | | 1 | | |
| Yes | 28/64 | 43.8 | 1.3 (0.75–2.1) | 0.4 | 1.4 (0.81–2.3) | | |
| Unknown | | | | | | | |
| Recent residence | | | | | | | |
| Karonga | 328/816 | 40.2 | 1 | | | 1 | 0.005 |
| Other Malawi | 56/176 | 31.8 | 0.69 (0.49–0.98) | | 0.58 (0.41–0.84) | 0.58 (0.40–0.84) | |
| Other country | 16/54 | 29.6 | 0.63 (0.34–1.1) | 0.04 | 0.48 (0.26–0.91) | 0.48 (0.26–0.91) | |
| Birth place | | | | | | | |
| Karonga | 267/659 | 40.5 | 1 | | 1 | | |
| Other Malawi | 81/227 | 35.7 | 0.81 (0.60–1.1) | | 0.79 (0.57–1.1) | | |

*Table 2. Continued on next page*

*Table 2. Continued*

| Characteristic | Linked/Total | | Association with links (unadjusted) | | Adjusted for age, sex, year, lineage | Adjusted for other variables included in model* | |
|---|---|---|---|---|---|---|---|
| | n/N | % | OR (95% CI) | p (lrtest) | OR (95% CI) | OR (95% CI) | p (lrtest) |
| Other country | 59/180 | 32.8 | 0.72 (0.51–1.0) | 0.1 | 0.67 (0.47–0.97) | | |

In this analysis individuals are defined as linked ('backwards links') using the cut-offs described in the text and if the closest link was with a patient within the previous 5 years. Extrapulmonary, recurrent cases, and cases before 1999 were excluded. Odds ratios (OR) calculated using logistic regression.

*In this model a dummy variable was used for the 32 individuals with missing data on recent residence.

†Test for trend.

Lineage-2 formed large clusters with small SNP differences. It had the highest proportion of disease due to recent transmission and the highest proportion of transmissions. Increased virulence in lineage-2 has been suggested previously (*Parwati et al., 2010*), often in association with drug resistance, but in this population all lineage-2 strains were drug sensitive. Despite these associations that suggest higher virulence and transmissibility, the proportion of cases due to lineage-2 did not increase over the period. We have previously reported that lineage-2 was first detected in this area in 1991, initially increased, and then plateaued from around 2000 (*Glynn et al., 2005a, 2010*). This may explain the lower proportion of lineage-2 strains in the oldest age group. The high proportion of linked cases with lineage-2 could reflect few imported (and therefore unlinked) cases, although there was no association between lineage and immigration.

In contrast, lineage-3 increased as a proportion of tuberculosis cases over time. It was associated with an intermediate proportion of disease due to recent infection and high transmission. In this population, it is also associated with relapse. Lineage-4 had smaller cluster sizes than lineages 2 and 3. It remains the most common lineage in this population, although the proportion has fallen over time.

Over the period of the study, the proportion of cases due to recent transmission decreased from 46% to 30%, and the proportion of cases transmitting and giving rise to new cases of tuberculosis also fell markedly. This correlates with a reduction in tuberculosis incidence over this period (*Mboma et al., 2013*). It suggests a considerable success of the tuberculosis and HIV control programmes, despite the potential for *M. tuberculosis* transmission in antiretroviral clinics.

We found no association with HIV infection in the proportion of disease due to recent infection (in contrast to our findings with RFLP in the earlier period [*Houben et al., 2009*, *2010*]) or in transmissibility. Social clustering of HIV-infected individuals may increase the opportunities for transmission to susceptible individuals who manifest disease, balancing out any decreased transmissibility. The change from our earlier findings could be due to the reduced transmission in the population and to the increasing use of isoniazid prophylaxis and antiretroviral therapy in HIV-positive individuals.

In this study, we had high quality whole genome sequence data on 72% of culture-positive patients over 15 years. While this is a high proportion, links will be missed, and the best link found may not be the correct one (especially when there are multiple patients with identical strains). Missing links will lead to underestimation of the proportion of disease due to recent transmission and of transmissions. The missing and wrongly attributed links are likely to be randomly distributed, leading to non-differential misclassification of linkage, and underestimation of associations with lineage and other factors.

This large, long-term study provides strong evidence for differences in transmission patterns and virulence between the *M. tuberculosis* lineages, particularly high transmissibility and virulence for lineages 2 and 3 and low transmissibility and virulence for lineage-1, which are unrelated to drug resistance, HIV infection, or host sub-population.

## Materials and methods

### Patients

In Karonga District, northern Malawi (population approximately 300,000), project staff at the hospital and peripheral health centres identify individuals with suspected tuberculosis (*Crampin et al., 2009*),

**Table 3**. Characteristics associated with transmissibility

| Characteristic | Any Linked/Total | | Association with links | | Adjusted for age, sex, year, lineage, smear status | |
| --- | --- | --- | --- | --- | --- | --- |
| | n/N | % | OR (95% CI) | p | OR (95% CI) | p (lrtest) |
| Overall | 431/1346 | 32.0 | | | | |
| Lineage | | | | | | |
| 1 | 59/217 | 27.2 | 0.87 (0.63–1.2) | | 0.94 (0.66–1.3) | |
| 2 | 27/61 | 44.3 | 1.7 (1.0–2.7) | | 1.9 (1.1–3.2) | |
| 3 | 65/154 | 42.2 | 1.6 (1.2–2.3) | | 1.9 (1.4–2.7) | |
| 4 | 280/914 | 30.6 | 1 | 0.006 | 1 | <0.001 |
| Age | | | | | | |
| <20 | 20/50 | 40.0 | 2.3 (1.2–4.4) | | 1.9 (0.98–3.7) | |
| 20–29 | 134/349 | 38.4 | 2.3 (1.5–3.3) | | 2.2 (1.5–3.3) | |
| 30–39 | 159/490 | 32.5 | 1.7 (1.2–2.5) | | 2.0 (1.3–2.9) | |
| 40–49 | 71/238 | 29.8 | 1.6 (1.0–2.4) | | 1.7 (1.1–2.7) | |
| 50+ | 47/219 | 21.5 | 1 | <0.001 | 1 | 0.002 |
| Sex | | | | | | |
| Female | 239/718 | 33.3 | 1 | | 1 | |
| Male | 192/628 | 30.6 | 0.87 (0.69–1.1) | 0.2 | 0.93 (0.73–1.2) | 0.5 |
| Year | | | | | | |
| 1995–1998 | 159/314 | 50.6 | 1 | | 1 | |
| 1999–2001 | 119/345 | 34.5 | 0.49 (0.36–0.66) | | 0.42 (0.31–0.58) | |
| 2002–2004 | 95/389 | 24.4 | 0.30 (0.22–0.41) | | 0.27 (0.19–0.37) | |
| 2005–2007 | 58/298 | 19.5 | 0.22 (0.16–0.32) | <0.001 | 0.20 (0.14–0.29) | <0.001 |
| TB type | | | | | | |
| Smear pos pulm | 338/1003 | 33.7 | 1 | | 1 | |
| Smear neg pulm | 93/343 | 27.1 | 0.72 (0.55–0.94) | 0.01 | 0.73 (0.55–0.96) | <0.001 |
| HIV status | | | | | | |
| HIV– | 91/318 | 28.6 | 1 | | 1 | |
| HIV+ no ART | 170/540 | 31.5 | 1.1 (0.83–1.5) | | 1.1 (0.81–1.6) | |
| HIV+ on ART | 11/48 | 22.9 | 0.70 (0.35–1.4) | 0.3 | 1.4 (0.62–3.1) | 0.6 |
| Previous TB | | | | | | |
| No | 391/1200 | 32.6 | 1 | | 1 | |
| Yes | 40/146 | 27.4 | 0.77 (0.53–1.1) | 0.2 | 0.85 (0.58–1.3) | 0.4 |
| INH resistance | | | | | | |
| No | 402/1237 | 32.5 | 1 | | 1 | |
| Yes | 29/100 | 29.0 | 0.86 (0.55–1.3) | 0.5 | 0.86 (0.54–1.4) | 0.5 |
| Recent residence | | | | | | |
| Karonga | 284/942 | 30.2 | 1 | | 1 | |
| Other Malawi | 80/234 | 34.2 | 1.2 (0.89–1.6) | | 1.0 (0.74–1.4) | |
| Other country | 20/74 | 27.0 | 0.88(0.52–1.5) | 0.4 | 0.57 (0.33–0.98) | 0.09 |
| Birth place | | | | | | |
| Karonga | 276/811 | 34.0 | 1 | | 1 | |
| Other Malawi | 80/272 | 29.4 | 0.83 (0.62–1.1) | | 0.82 (0.60–1.1) | |

*Table 3. Continued*

| Characteristic | Any Linked/Total | | Association with links | | Adjusted for age, sex, year, lineage, smear status | |
|---|---|---|---|---|---|---|
| | n/N | % | OR (95% CI) | p | OR (95% CI) | p (lrtest) |
| Other country | 64/234 | 27.4 | 0.77 (0.56–1.1) | 0.2 | 0.71 (0.51–0.99) | 0.08 |

The numbers of likely transmissions ('forward links') were compared by individual characteristics using ordered logistic regression. Extrapulmonary cases and cases occurring after 2007 were excluded.

and sputum and other specimens are taken. All diagnosed tuberculosis patients are interviewed, and HIV-tested, after counselling and if consent is given. The incidence of new smear-positive tuberculosis in adults in the district has fallen from 124/100000/year to 87/100000/year over the period of this study, with about 6% isoniazid resistance and <1% multidrug resistance (*Mboma et al., 2013*). Adult HIV prevalence in the area is around 10%.

Approval for the study was given by the ethics committee of the London School of Hygiene & Tropical Medicine (#5067) and the Malawian National Health Sciences Research Committee (#424). Informed consent was obtained from all participants.

## Cultures and sequencing

Culture is performed in the project laboratories in Malawi, with species identification and drug susceptibility testing in the UK Mycobacterium Reference Laboratory (*Mboma et al., 2013*). RFLP was performed on cultures from all patients from late 1995–2008 (*Glynn et al., 2005b*). We processed all available stored DNA samples or cultures from 1995 to 2010 for whole genome sequencing at the Sanger Institute, using Illumina HiSeq 2000, paired-end reads of length 100 base-pairs.

## Read quality filtering

We used trimmomatic software (http://www.usadellab.org/cms/?page=trimmomatic) to remove low-quality reads and low-quality 3′ ends of reads, keeping only reads ≥50 base-pairs long, with nucleotides >Q27 (equivalent to a risk of error of <0.2% per read per base-pair).

We mapped reads for each sample against the H37Rv reference genome (Genbank assession: AL123456.3), using the *BWA-mem* algorithm (http://bio-bwa.sourceforge.net/) (*Li, 2013*). We excluded samples with average genomic coverage less than 10-fold.

We identified SNP positions using *SAMtools* (http://samtools.sourceforge.net/) (*Li et al., 2009*). Sample genotypes were called using the majority allele (minimum frequency 75%) in positions supported by at least 20-fold coverage; otherwise we classified them as missing (thus ignoring heterozygous calls). We excluded samples with >15% missing genotype calls, to remove possible contaminated or mixed samples or technical errors. (The proportion of mixed strains is low in this setting [*Mallard et al., 2010*]). We excluded genome positions with >15% missing genotypes, and those in highly repetitive and variable regions (e.g., PE/PPE genes).

In the final analysis, 94% of the *M. tuberculosis* genome was analysed for variants. Median coverage was 88-fold, mean 127. Spoligotyping was performed in silico using SpolPred (*Coll et al., 2012*). Lineages were defined from spoligotype families (*Demay et al., 2012*).

We calculated SNP distances between sequences using the ape library in the R statistical package (http://cran.r-project.org/). We computed a maximum-likelihood phylogenetic tree including all samples, using RAxML, using the GTRCAT model.

## Transmission mapping

For the transmission network, we used the SeqTrack package in R (*Jombart et al., 2011*), using one sample per person-episode of disease, excluding episodes of extrapulmonary tuberculosis (as these cannot transmit). This builds a minimum-spanning tree, minimizing the genomic distance between links and keeping the disease onset dates coherent. Based on our data, we allowed up to 10 SNPs difference for inclusion in the networks. The suitability of the cut-off was assessed by examination

of the SNP differences within and between patients. We have previously shown in 92 patients in this data set with repeat samples from the same or different episodes of disease that, using the same pipeline, there is a clear bimodal distribution, with pairs of samples either having up to 8 SNPs between them or more than 100 SNPs (*Guerra-Assuncao et al., 2014*). Furthermore, among 187 pairs of individuals with epidemiological links, 62 had ≤10 SNPs, 9 had 10–99 SNPs, and 116 had ≥100 SNPs.

Statistical analysis used STATA 13 (http://www.stata.com/). We estimated the between-patient mutation rate using linear regression of the number of SNPs by time between disease onset dates (taken as the date of first evidence of tuberculosis—the earliest of date of collection of the first positive sample, or registration or treatment) in the patients connected in the network. This analysis was repeated using dates of specimen collection. For comparison, we calculated the within-patient mutation rate, in individuals with more than one specimen from the same episode of disease or from a relapse, also using a cut-off of ≤10 SNPs.

We compared the size of the clusters by lineage. Among the likely transmissions (≤10 SNPs), we examined the number of SNP differences and mutation rates by lineage, and characteristics of the index and subsequent case, using non-parametric tests (Wilcoxon rank-sum and the equality-of-medians test).

For the analyses of risk factors for disease due to recently acquired infection and for transmissibility, we classified links with 6–10 SNPs different as uncertain unless the mutation rate was <0.003 SNPs/day, to allow for larger changes over long time periods. Those patients with uncertain links were excluded from the risk factor analyses.

## Disease due to recently acquired infection

The SeqTrack network shows the most likely source of infection for each case. For groups of cases with zero SNPs between them the one closest in date was chosen. A case was defined as due to recently acquired infection if the most likely source was within 5 years, and not due to recent infection if there was no source identified or if the source was earlier than this (even if there were other closely related strains within 5 years).

In this analysis of 'backwards' links, we used the first 3 years of data only for identifying previous links. We examined risk factors for disease due to recent infection among individuals with their first episode of tuberculosis, using logistic regression. The multivariable analysis included lineage, age, sex, and year a priori, and other factors if they were associated with recent infection after adjustment for these, or if they confounded other variables.

## Transmissibility

The SeqTrack network links can also be used to examine forward transmission that results in disease. In this analysis, we used the last 3 years of data only to identify transmissions that had taken place, to allow time for transmissions causing new cases. We used ordered logistic regression to assess risk factors for transmission and the number of transmissions. In the multivariable analysis, we adjusted for lineage, age, sex, year, and sputum smear status of the index case a priori, and assessed confounding by other factors. We repeated the analysis using logistic regression, comparing any transmissions vs none. Since those in later years had less time for transmission to be detected, we examined the effect of calendar period using transmission within 3 years of the index case.

## Repositories for data and software

Software sources for in-house programs will be made available on sourceforge.net (http://sourceforge.net/projects/patogenico/). Raw data can be obtained from the European Nucleotide Archive at EMBL-EBI (project accessions: ERP000436 and ERP001072).

## Acknowledgements

We thank the Government of the Republic of Malawi for their interest in this Project and the National Health Sciences Research Committee of Malawi for permission to publish the paper. We thank the Wellcome Trust Sanger Institute core and pathogen sequencing and informatics teams.

## Additional information

### Funding

| Funder | Grant reference number | Author |
|---|---|---|
| Wellcome Trust | 096249/Z/11/B | AC Crampin, R McNerney, J Parkhill, TG Clark, JR Glynn |

The funder had no role in study design, data collection and interpretation, or the decision to submit the work for publication.

### Author contributions

JG-A, TGC, Conception and design, Analysis and interpretation of data, Drafting or revising the article; ACC, RH, Conception and design, Acquisition of data, Drafting or revising the article; TM, KM, PK, LB, AC, RP, RMN, JP, Acquisition of data, Drafting or revising the article; FC, Analysis and interpretation of data, Drafting or revising the article; PF, Conception and design, Drafting or revising the article; JRG, Conception and design, Acquisition of data, Analysis and interpretation of data, Drafting or revising the article

### Author ORCIDs

T Mzembe, http://orcid.org/0000-0002-3998-3951

### Ethics

Human subjects: Approval for the study was given by the ethics committee of the London School of Hygiene & Tropical Medicine (#5067) and the Malawian National Health Sciences Research Committee (#424). Informed consent was obtained from all participants.

## Additional files

### Major datasets

The following datasets were generated:

| Author(s) | Year | Dataset title | Dataset ID and/or URL | Database, license, and accessibility information |
|---|---|---|---|---|
| Guerra-Assunção JA, Crampin AC, Houben RMGJ, Mzembe T, Mallard K, Coll F, Khan P, Banda L, Chiwaya A, Pereira RPA, McNerney R, Fine PEM, Parkhill J, Clark TG, Glynn JR | 2010 | Whole_genome_sequencing_of_*Mycobacterium_tuberculosis*_from_a_population_in_Malawi___transmission_dynamics_and_associations_with_HIV | http://www.ebi.ac.uk/ena/data/view/ERP000436 | Publically available at the EBI European Nucleotide Archive (http://www.ebi.ac.uk/ena). |
| Guerra-Assunção JA, Crampin AC, Houben RMGJ, Mzembe T, Mallard K, Coll F, Khan P, Banda L, Chiwaya A, Pereira RPA, McNerney R, Fine PEM, Parkhill J, Clark TG, Glynn JR | 2011 | Karonga_Prevention_Study__KPS___whole_genome_sequencing_in_a_whole_population | http://www.ebi.ac.uk/ena/data/view/ERP001072 | Publically available at the EBI European Nucleotide Archive (http://www.ebi.ac.uk/ena). |

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
