## [Decision Letter]

Thank you for sending your work entitled “Large scale whole genome sequencing of
*M. tuberculosis* provides insights into transmission in a high
prevalence area” for consideration at *eLife*. Your article has
been favorably evaluated by Prabhat Jha (Senior editor), a Reviewing editor, and two
reviewers.

The Reviewing editor and the reviewers discussed their comments before we reached this
decision, and the Reviewing editor has assembled the following comments to help you
prepare a revised submission.

Kindly elaborate the analysis plan in relation to unraveling the transmission clusters,
e.g. were there a priori hypothesis and criteria set out or was this determined in an ad
hoc manner? How are epidemiological transmission clusters related to the clusters from
the RFLP analysis?

*Reviewer #1*:

This manuscript reports the results of a large-scale whole genome sequencing analysis of
*M. tuberculosis* isolates obtained in a population-based study in
Malawi spanning 15 years. The study is unique in its scope and size. The analyses
provide important insights in transmission dynamics, in particular in relation to major
lineages. The manuscript is well written, the analyses are sound and the tables and
figures are clear. In summary, an important contribution to the subject that deserves
publication provided that some (overall rather minor) concerns are addressed.

From Figure 2 it seems that there are different
individuals with isolates within the same RFLP that have up to 50 SNPs difference.
Authors took a rather arbitrary cut-off for distinguishing links of up to 10 SNPs
difference. It would be important to see to what extent their main findings are
sensitive to this choice of cut-off, in particular the associations with, and
differences in mutation rates for lineages. This could be done by repeating the main
analyses at a higher cut-off.

In the Methods section: “We estimated the between-patient mutation rate
(…) between disease onset dates (…)”. How were disease onset dates
ascertained? Were diagnostic delays estimated and if so, how? How robust are these
estimates? I would suspect that dates of onset are associated with date of diagnosis
with certain elasticity, as will be the dates of specimen collection. Therefore
predictor estimates may be confounded by determinants of (reported) delay. I am not sure
that the multivariable analyses take this confounding sufficiently into account, and
authors should acknowledge this potential shortcoming and its consequences for their
conclusions in the Discussion section.

*Reviewer #2*:

The authors should be commended for this excellent study which describes the largest
collection of whole genome sequences of *M. tuberculosis* isolates
assembled to date. The central aim of the study was to use whole genome sequences to
construct a transmission network of cases resident in a high prevalence setting. This is
a challenging exercise as the complexity of transmission networks increases with
incidence. The authors also used their data to estimate the evolutionary rate of the
*M. tuberculosis* genome and the proportion of recent transmission
events. This study extends the current knowledge of transmission and genetic variability
occurring in *M. tuberculosis* isolates from a high incidence
setting.

The hypothesis that genetic distance is a measure of transmission is not validated
within this setting: the authors should provide additional data to support the notion
that isolates differing by less than 10 SNPs reflect transmission.

---

## [Author Response]

*Kindly elaborate the analysis plan in relation to unraveling the transmission
clusters*, *e.g. were there a priori hypothesis and criteria set out
or was this determined in an ad hoc manner? How are epidemiological transmission
clusters related to the clusters from the RFLP analysis?*

We have addressed these points below, where they are raised by the reviewers.

*Full major concerns*:

Reviewer #1:

*From*
Figure 2
*it seems that there are different individuals with isolates within the same RFLP
that have up to 50 SNPs difference. Authors took a rather arbitrary cut-off for
distinguishing links of up to 10 SNPs difference. It would be important to see to
what extent their main findings are sensitive to this choice of cut-off, in
particular the associations with, and differences in mutation rates for lineages.
This could be done by repeating the main analyses at a higher cut-off*.

We have added further justification for our cut-off to the Methods section: “We
have previously shown in 92 patients in this dataset with repeat samples from the same
or different episodes of disease that, using the same pipeline, there is a clear bimodal
distribution, with pairs of samples either having up to 8 SNPs between them or more than
100 SNPs (16).
Furthermore, among 170 pairs of individuals with epidemiological links, 62 had
≤10 SNPs, 9 had 10-99 SNPs and 116 had ≥100 SNPs.”

Other studies have used a similar or stricter cut-off. To look for associations with
transmission we were keen to have a cut-off with high specificity.

*In the Methods section: “We estimated the between-patient mutation rate
(…) between disease onset dates (…)”. How were disease onset
dates ascertained? Were diagnostic delays estimated and if so, how? How robust are
these estimates? I would suspect that dates of onset are associated with date of
diagnosis with certain elasticity, as will be the dates of specimen collection.
Therefore predictor estimates may be confounded by determinants of (reported) delay.
I am not sure that the multivariable analyses take this confounding sufficiently into
account, and authors should acknowledge this potential shortcoming and its
consequences for their conclusions in the Discussion section*.

We have clarified that we used the date of first evidence of tuberculosis. (We have not
attempted to estimate diagnostic delay). We have added this to the Discussion.

Reviewer #2:

*The hypothesis that genetic distance is a measure of transmission is not
validated within this setting: the authors should provide additional data to support
the notion that isolates differing by less than 10 SNPs reflect
transmission*.

See our response above to reviewer 1. We have now added further data to justify this
decision, based on repeat specimens from individuals and on comparisons between
individuals with known epidemiological links.